# Neurospecific Molecules Measured in Periphery in Humans: How Do They Correlate with the Brain Levels? A Systematic Review

**DOI:** 10.3390/ijms23169193

**Published:** 2022-08-16

**Authors:** Maria A. Tikhonova, Svetlana Y. Zhanaeva, Anna A. Shvaikovskaya, Nikita M. Olkov, Lyubomir I. Aftanas, Konstantin V. Danilenko

**Affiliations:** Scientific Research Institute of Neurosciences and Medicine (SRINM), 630117 Novosibirsk, Russia

**Keywords:** human, brain, plasma, serum, blood cells, concordance, post-mortem, neuroimaging, immunoassay, blood-brain barrier

## Abstract

Human brain state is usually estimated by brain-specific substances in peripheral tissues, but, for most analytes, a concordance between their content in the brain and periphery is unclear. In this systematic review, we summarized the investigated correlations in humans. PubMed was searched up to June 2022. We included studies measuring the same endogenous neurospecific analytes in the central nervous system and periphery in the same subjects. Not eligible were studies of cerebrospinal fluid, with significant blood–brain barrier disruption, of molecules with well-established blood-periphery concordance or measured in brain tumors. Seventeen studies were eligible. Four studies did not report on correlation and four revealed no significant correlation. Four molecules were examined twice. For BDNF, there was no correlation in both studies. For phenylalanine, glutamine, and glutamate, results were contradictory. Strong correlations were found for free tryptophan (*r* = 0.97) and translocator protein (*r* = 0.90). Thus, only for three molecules was there some certainty. BDNF in plasma or serum does not reflect brain content, whereas free tryptophan (in plasma) and translocator protein (in blood cells) can serve as peripheral biomarkers. We expect a breakthrough in the field with advanced in vivo metabolomic analyses, neuroimaging techniques, and blood assays for exosomes of brain origin.

## 1. Introduction

The identification of reliable biomarkers is a key approach to the early diagnosis of pathological processes in the organism. Moreover, it also contributes to a better understanding of normal neurophysiology and pathogenetic mechanisms of neuropsychiatric diseases. As for other organs and systems, features and state of the human brain are usually evaluated by analyzing brain-specific substances in peripheral biological fluids that are easily accessible for research, i.e., blood, urine, saliva, and, less often, cerebrospinal fluid (CSF). However, due to the blood-brain barrier (BBB) and a number of other reasons, such correlations or associations may be weak or completely absent. The relation between the concentration of neurospecific molecules in the periphery and their content in the brain remains unknown for most analytes. The exceptions appear to be plasma amyloid-β and tau protein, which have been found to tightly reflect their brain levels and are candidates for inclusion in algorithms for early diagnostics of Alzheimer’s disease [1].

### 1.1. Neurospecific Proteins

Brain tissue consists of two basic classes of specialized cells: neurons and glial cells. The latter act to support both the structure and function of neurons [2]. Hence, the brain-specific analytes could be defined as endogenous molecules that are secreted or expressed in the brain and are associated with the functioning of the nervous tissue; namely, neuron-derived substances related to neurotransmission (e.g., neurotransmitters, receptors, enzymes, transporters, and neuropeptides) and glia-derived molecules, including neurotrophic factors. Microglial cells are the resident immune cells in the brain and, therefore, should be specifically mentioned. Microglia account for 5–20% of the total glial cell population within the brain parenchyma and share origin and main features with peripheral macrophages. Nevertheless, in addition to immune functions, emerging reports support their fundamental role in the control of the local microenvironment, providing neuronal proliferation and differentiation, as well as the formation of synaptic connections [3]. Thus, the molecules produced by microglia for the maintenance and functioning of neurons can be regarded as the brain-specific analytes.

### 1.2. Blood–Brain Barrier (BBB)

The transport of molecules to and from the brain is controlled by the BBB. The BBB is a unique anatomical and physiological dynamic interface between the peripheral blood supply and the brain with its CSF. Tight junctions between the endothelial cells of the cerebral microvasculature limit the passage of large, negatively charged molecules via paracellular diffusion, whereas transcellular transportation across the endothelial cell is controlled by a number of mechanisms, including transporter proteins, endocytosis, and diffusion [4,5]. Large compounds, such as brain-specific proteins, do not usually cross the BBB; for some of them, the permeability is still unknown. For example, in humans, it is not consensual yet whether brain-derived neurotrophic factor (BDNF) easily permeates the BBB (see references in [6]), and, therefore, whether blood BDNF may reflect central BDNF.

A disruption in BBB integrity may lead to BBB dysfunction and increase in vascular permeability that allows protein leakage in both directions. Conditions in which an increased BBB permeability of various extents have been reported include coma, brain trauma, ischemia, infection, hypertension, psychiatric and neurodegenerative disorders [7,8,9]. For instance, S100B, neuron-specific enolase (NSE), and some other neurospecific analytes are widely used in clinical practice as serum biomarkers of traumatic brain injury; they help to predict severity of injury and intracranial disease burden, and to assess brain damage and clinical outcome [10,11,12]. Hence, we did not consider brain injury and prolonged coma in the present review.

On the other hand, some neurospecific molecules are synthesized in peripheral tissues and cells. For example, BDNF in humans is widely expressed outside of the central nervous system (CNS), namely, in white blood cells, platelets, vascular endothelium, smooth muscles, and other peripheral tissues [13]. Peripheral blood mononuclear cells (PBMCs) express many CNS enzymes, receptors, and downstream signaling proteins. Moreover, in several neuropsychiatric disorders, alteration of metabolism and cellular functions in the CNS are concomitant with changes in blood lymphocytes [14,15]. Hence, PBMCs are considered to be a suitable cellular model for drug discovery and studying the mechanisms of neuropsychiatric disorders [16].

### 1.3. Post-Mortem versus In Vivo

Analytes in the human brain could be measured post-mortem, as well as in vivo, following brain surgery or using neuroimaging techniques. Some surgeries include resection of brain tissue, e.g., removal of brain tumors, epileptic lesions, lobotomy, and tractotomy. The latter two were used in earlier years to treat mental disorders. The brain visualization techniques for measuring analytes are magnetic resonance imaging (MRI) complemented with magnetic resonance spectroscopy (MRS), positron-emission tomography (PET), and single-photon emission computed tomography (SPECT). Compared to in vivo studies, the validity of the post-mortem brain measurements may be compromised by proteolytic degradation of some molecules with the time elapsed from death to sample conservation [17]. An additional variability may occur because of post-mortem stability of some proteins and susceptibility to post-mortem degradation of the others [18]. Another disadvantage of the post-mortem sampling is that it is usually performed following a terminal phase of the disease when the permeability of the BBB is compromised and, thus, blood and brain concentrations may mirror each other due to the leakage.

### 1.4. Measurements

The measurement in the brain, blood, and CSF engages various biochemical methods. A significant number of studies use optical enzyme-linked immunoassay technologies, such as ELISA, and Luminex xMAP, while, in the last three decades, electrochemiluminescence-based (ECL) immunoassay and single-molecule array (Simoa, also called digital ELISA) have appeared in the set of immunoassay methods. In these technologies, a detectable light signal (following light inflection in ELISA or fluorescence for xMAP) is generated as a result of the binding of a ligand (analyte) to specially prepared antibodies. Luminex xMAP has greater sensitivity and specificity compared to classical ELISA and can simultaneously assay many analytes in a single sample. However, their sensitivity is insufficient for measuring some brain-specific proteins in blood, e.g., neurofilament light (NfL) and glial fibrillary acidic protein (GFAP), due to their low concentration (pg/mL) [19]. Simoa is an “advanced” xMAP immunoassay isolating and detecting single immune complexes which can measure proteins at subfemtomolar concentrations [20,21,22]. The analytical sensitivity of ECL and Simoa methods is higher compared to traditional ELISA by ~10 and ~1000 times, respectively. In recent years, other ultrasensitive optical technologies have been developed, such as Surface plasmon resonance (SPR) and Surface-enhanced Raman spectroscopy (SERS), which allow real-time assay of femtomolar concentrations of label-free analytes [23,24].

Immunohistochemical and immunocytochemical analyses are close to the above-described enzyme-linked immunoassay technologies. They are also based on the binding between a ligand and specific antibodies. The resulting signal represents immunofluorescence or chromogenic immunohistochemistry associated with a marker expression and is usually detected using microscopy. The methods provide great visual options, including spatial distribution of a marker in a tissue section or a marker expression in specific cells or cell compartments, as well as marker co-expression [25,26]. The main limitation of the approach is the semi-quantitative nature of its outcomes. The data are expressed as arbitrary units and could not be calibrated using external standards as in enzyme-linked immunoassays.

Mass spectrometry (MS), coupled with liquid chromatography (LC), is an in vitro method during which small (<3 kDa) molecules of a biosample are passed sequentially in liquid (or, more rarely, in gas) through a chromatographic column, charged by electron ionization or contact with acidic water or some other way, and further differentiated by a mass-spectrometer according to their mass-to-charge ratio. Identification of the metabolites is performed by comparison of the obtained mass spectrum (with each peak representing a molecule or its fragment) with a reference compound library or a database [27]. Up to 1000–2500 molecules can be measured simultaneously, albeit not all molecules could be identified by LC/MS. The method allows comparison of metabolomic and proteomic profiles in different diseases [28,29].

Nuclear magnetic resonance spectroscopy (NMR or MRS) is an alternative method, also used in many studies. A biosample is put in a strong constant magnetic field where nuclear magnetic spins are first aligned and then perturbed by weak radio frequency pulses, generating detectable electromagnetic signals. The latter depend on the magnetic properties of a target isotope normally present in tissues (^1^H commonly) and chemical environment. The resulting signals are then converted into NMR spectra. While this method does not require chromatographic steps and provides the possibility to reuse a sample after the measurement, its main disadvantage, compared to MS, is a lower sensitivity and selectivity. MRS coupled with MRI allows in vivo measurement of certain metabolites including some neurospecific ones, the number of which is currently limited to ~5–21, depending on the target isotope and magnet power. For example, spectral peaks for glutamine and glutamate are distinguishable from each other only with a ≥3 Tesla magnet [30]. Metabolite concentration can be measured using ^1^H-MRS/MRI by normalizing signal intensity in the spectra to the signal from creatine.

Scintigraphic quantification of radioisotopes distributed in the body after their infusion into the blood is the essence of PET and SPECT (gamma-emission), usually coupled with computer tomography (CT) or MRI. Both are increasingly used to estimate energy metabolism, blood flow, and chemical absorption (e.g., binding of brain receptors or transporters) but may also estimate a specific brain chemical (which is relevant to this review) in the case of development of an appropriate radioactive tracer [31,32]. While PET has the advantage of higher resolution and sensitivity, SPECT is more accessible and cheaper.

Genetic (e.g., polymorphisms or functional mutations) and epigenetic (e.g., methylation) information, as well as different regulators of transcription (e.g., microRNAs), have an unquestionable impact on protein synthesis through the gene expression. Nevertheless, there is a certain gap between mRNA levels and protein content that makes research of the concordance of these parameters between the brain and periphery even more sophisticated. Protein and mRNA levels do not necessarily match, especially if posttranscriptional modifications occur and play an essential role [33]. Considering methylomic profiling of whole blood versus brain tissue, for the majority of DNA methylation sites, interindividual variation in whole blood is not a strong predictor of interindividual variation in the brain [34]. Thus, mRNA expression and DNA methylation patterns were out of the scope of the review.

### 1.5. CSF—Not Considered

CSF studies were not considered for the current review, since, in practice, CSF is poorly accessible for examination. Besides, it can only conditionally be considered peripheral, since it lies along with the brain on one side of the BBB. There are numerous original and review studies that have examined the CSF-blood correlation for certain analytes. One of the most impressive is a recent metabolomic study by Rogachev et al. (2021) [35] that identified as many as 101 analytes in the samples of both CSF and plasma collected on the same day in glioma patients and healthy controls. The statistical analysis showed a significant correlation between plasma and CSF for the majority of metabolites. This, however, does not mean that brain analytes are reflected in the CSF and blood similarly. For example, plasma GFAP was found to discriminate amyloid-β-positive from amyloid-β-negative individuals more accurately than CSF GFAP [36].

### 1.6. Study Objectives

To our knowledge, systematic analysis of concordance between the levels of neurospecific substances in the brain and peripheral tissues in humans has not been performed yet, although this topic was partially addressed for some molecules, e.g., a recent review on the correlation between peripheral and central kynurenine metabolite concentrations in psychiatric disorders [37]. We present a systematic review of the studies investigating brain-specific analytes measured both in the brain and in the periphery in the same individuals in vivo or post-mortem with the aim to analyze the reported correlation between peripheral and central measures of neurospecific endogenous substances. We then discuss the main approaches and methods applied, along with the difficulties, challenges, and prospects.

## 2. Methods

The review was conducted and reported according to the PRISMA guidelines [38]. The work explored three sources for retrieving eligible articles: articles that authors already knew, PubMed library, and citation searching. The eligibility criteria were as follows: (1) brain and periphery measurements must consider the same subjects and the same analytes; (2) the analytes are endogenous molecules predominantly associated with the functioning of the nervous tissue (e.g., not glucose used at scintigraphic neuroimaging, lipids, microelements, or exogenous substances). The following were not eligible: (3) animal studies; (4) CSF studies; (5) studies of molecules with already well-established blood-periphery concordance (e.g., tau protein, amyloid-β peptide); (6) pathological conditions characterized by significant BBB disruption (due to [fatal] brain injury, prolonged ventilation or coma); (7) analytes measured in brain tumors (e.g., glioblastoma-specific).

The search for eligible articles in PubMed was performed from the first reports published in October 1970 through 23 June 2022. The search string was constructed in such a way that it reduced the number of identifiable records to a manageable few thousand without significantly compromising coverage, for which the script was tested on 29 publications of interest already known to the authors from their previous scientific activity. The search string was as follows: (“chemical” OR “chemicals” OR “molecule*” OR “protein” OR “peptide” OR “substance” OR “analyte” OR “metabolite” OR “acid” OR amine OR “biomarker” OR “metabolom*” OR “proteom*”) AND (“concentration*” OR “level*” OR “content*” OR “activity” OR “rate*” OR “expression” OR “AUC”) AND (“brain” OR “hippocamp*” OR “cortex” OR “lobe” OR “amygdala” OR “glia”) AND (“blood” OR “serum” OR “plasma” OR “leukocytes” OR “mononuclear” OR “thrombocytes” OR “erythrocytes” OR “nasal” OR “olfactor*” OR “endothelial cell” OR “muscle”) AND (“post-mortem” OR postmortem OR “died” OR “dead” OR “brains” OR surgery OR “resect*” OR “lobotom*” OR “tractotom*” OR “PET” OR “positron emission tomography” OR “SPECT” OR “spectroscopy”) AND (correlat* OR associat* OR concord*) NOT (review [Publication Type]) NOT (“Case reports” [Publication Type]) NOT (“brain natriuretic peptide”). The filter “humans” was used. The PubMed search was limited to titles when the papers lacked an abstract.

The PubMed search generated 5642 records. They were divided equally between five authors for individual assessment. Scanning of titles and abstracts resulted in identification of 98 articles for full-text scrutiny. An additional search was performed by analyzing relevant citations met in the following: (1) in each scrutinized article, especially eligible ones, (2) in the list of other articles which cited an eligible article (according to scholar.google.com), and (3) in occasionally viewed review articles. The citation search brought eight additional articles to scrutiny. In case of doubt regarding eligibility, the full-text article was assessed collectively to get a decision on its inclusion in the review. Correlation between the analytes was termed as weak, moderate, or strong according to the correlation coefficient (*r* for Pearson or *rho* for Spearman statistics) of 0.3–0.5, 0.5–0.7, or >0.7, respectively (absolute values).

## 3. Results and Discussion

Seventeen studies were considered to be eligible (14 were among the PubMed records, three found during the citation search; Figure 1). Data are presented in Table 1. Four of these studies did not report on brain–periphery correlation (including both metabolomic studies). Nevertheless, they were included in Table 1 to see their potential to modify the systematic review’s conclusion if these correlations were reported in future publications.

The studies were heterogeneous with respect to all analyzed variables (see column names in Table 1), except that periphery was represented exclusively by blood specimens: whole blood (N = 3), plasma (N = 8), serum (N = 4), or blood cells (N = 2). The studied populations encompassed patients with Alzheimer’s disease (N = 5), Parkinson’s disease (N = 1), mood disorders (N = 3), epilepsy (N = 1), non-neuropsychiatric illnesses (N = 4), healthy subjects (N = 3) and control groups. In total, 258 subjects were used in the correlation analysis. Eight brain studies were post-mortem studies and there were nine in vivo studies. In post-mortem studies, the peripheral specimens were often sampled not on the same day of the brain sampling but years before death (N = 5), whereas in in vivo studies, it was the case in only one study [39] (a within-week difference). Measurement techniques encompassed various biochemical methods (all studies), MRS (N = 3), PET (N = 1), SPECT (N = 1), and immunocytochemistry or immunohistochemistry (N = 2).

### 3.1. Small Molecules

Nine studies investigated small molecules (Table 1). Of these nine studies, two measured quinolinic acid, a neurotoxic tryptophan-kynurenine pathway metabolite, NMDA agonist. One reported no significant brain-serum correlation [40], and another one, unfortunately, did not report on correlation [41]. One early study measured another tryptophan-kynurenine-related metabolite, tryptophan (a serotonin precursor), and found very strong cortex-plasma correlation for free tryptophan and a moderate one for total tryptophan [42,43].

Seventeen smaller amino acids were biochemically measured in a study by Honig et al. (1998) [44], and none significantly correlated between the brain cortex and plasma (or CSF; Table 1). Two of these amino acids were separately measured later in two studies, each of which used MRS for the brain examination. Phenylalanine was measured in a study by Koch and colleagues (2000) [45]. Calculations, which we made using raw phenylalanine values presented in the article, indeed yielded a moderate and significant brain-blood correlation. Glutamine (and also glutamate) was measured in a recent study by Takado et al. (2019) [30]. A strong positive cortex-plasma correlation was found, whereas for glutamate such correlation was absent. The study also demonstrated that the correlation may be brain-area-specific. Contrary to the latter study, Shulman and colleagues (2006) [39] did not find a correlation for glutamate, probably because “blood sampling and MRS acquisition were performed on different days” [30].

A big set of small molecules (143 and 129) was measured in two post-mortem metabolomic studies [46,47]. Interestingly, both used the same banking sources for specimen acquisition and published their results in the same year. Wang and colleagues (2020) [47] analyzed all 17 amino acids mentioned in the study by Honig et al. (1988) [44] (Table 1), but not quinolinic acid, tryptophan, or glutamate (measured in other studies included in Table 1). The majority of the molecules in the analysis were in fact not neurospecific. However, the article did not report on brain-serum correlation for single molecules. The same is true for the article by Huo and colleagues (2020) [46], who did not even specify the molecules studied.

### 3.2. Peptides

Eight studies investigated peptides, molecules built of amino acids (Table 1). Three of them measured neurotrophic factors: BDNF, pro-BDNF, GDNF (glial cell line-derived neurotrophic factor), or NGF (nerve growth factor). One did not report on correlation. Another one found a weak hippocampus-serum correlation for pro-BDNF but not for BDNF or the other two brain areas [48]. The third one also did not find a correlation for BDNF, but found a trend for GDNF [6].

Another widely studied analyte NfL, an axon-specific protein, was found to be correlated between post-mortem brain and plasma collected at the closest time to death, albeit the strength of the correlation was again weak [49].

The next study analyzed CREB, acyclic-AMP response element binding protein (which regulates transcription of genes of several neurospecific molecules, including BDNF, tyrosine hydroxylase) and three CREB-related molecules, namely, pCREB, CBP and p300 (Table 1; [50]). The analytes were assayed biochemically in the prefrontal cortex and PBMCs. One brain-PBMC correlation was found to be significant (for pCREB) but only for a subgroup of Alzheimer’s disease patients whose blood was taken <3 years before death.

In an in vivo study of Parkinson’s patients, the dopamine transporter protein availability in the striatum and its content in blood lymphocytes did not correlate, probably because of the different methods applied (SPECT for putamen and immunocytochemistry for lymphocytes; [51]). Contrary to the negative result in that study, a strong correlation was found for translocator protein (TPRO), belonging to a family of tryptophan-rich sensory proteins. It is localized in mitochondrial membranes and provides transport of cholesterol and some other molecules into mitochondria in the body. TPRO is studied mainly in relation to the nervous/immune system. The TRPO content was determined by PET for both the whole brain and circulating blood cells [52].

In another in vivo study, acetylcholinesterase activity in brain gliomas and whole blood was highly intercorrelated, decreasing sharply in both specimens in the grade I to grade IV tumor groups [53]. Although this study did not meet review criterion #7 (“tumor studies not included”), the major argument for inclusion was that in patients with the lowest grade glioma (I), the levels of the acetylcholinesterase activity were similar to the reference values in the brain and blood obtained from healthy subjects [53].

Contrary to the metabolomic studies of small molecules, for peptides, there were no proteomic studies eligible for the review.

**Table 1 ijms-23-09193-t001:** Eligible studies.

Study	Subjects in Correlation Analysis	Conditions ^	Measures, Specimens and Techniques	Results
**Small Molecules**				
Heyes et al., 1998 [40]	16 AIDS (acquired immunodeficiency syndrome) patients	p-m +	Quinolinic acid: brain (basal ganglia, cortical white matter, cortical gray matter) vs. serum (and CSF), by chemical ionization-gas chromatography	No significant brain-serum (and brain-CSF) correlations
Basile et al., 1995 [41]	58 patients with liver failure and encephalopathy, 18 normal subjects	p-m −	Quinolinic acid: brain vs. plasma (taken before death), by MS	No report on correlation
Gillman et al., either 1980 or 1981 [42,43]	5 psychiatric patients during tractotomy (who were not tryptophan-infused)	i-v +	Total and free tryptophan: brain cortex, by high performance liquid chromatography vs. plasma (and CSF), using “Chromaspek amino acid analyser”	Prominent cortex-plasma correlations: for total tryptophan *r* = 0.58 (ns), for free tryptophan ***r* = 0.97 (*p* < 0.01)**
Honig et al., 1988 [44]	14 patients with refractory depression during tractotomy	i-v +	17 amino acids (taurine, asparagine, threonine, serine, glutamic acid, glutamine, glycine, alanine, valine, methionine, isoleucine, leucine, tyrosine, phenylalanine, histidine, lysine, arginine): brain cortex vs. plasma (and CSF), using “Chromaspek amino acid analyser”	No significant brain-plasma (or brain-CSF) correlations for all amino acids (except for gamma-aminobutyric acid GABA, which was undetectable in plasma and CSF)
Koch et al., 2000 [45]	4 subjects with phenylketonuria and 5 healthy controls	i-v +	Phenylalanine: brain (by MRI/MRS) vs. blood (by amino acid analyzer)	Significant brain-blood correlation ***rho* = 0.51 (*p* < 0.05)**
Takado et al., 2019 [30]	19 healthy subjects	i-v +	Glutamine and glutamate: brain posterior cingulate cortex (PCC) and cerebellum, by photon magnetic resonance spectroscopy (MRI/MRS; twice within 1 h) vs. plasma (taken once between the two MRS sessions), by LC/MS)	Significant brain PCC-plasma correlation for glutamine (mean of two measurements) ***rho* = 0.72 (*p* < 0.01)**. No other correlations significant
Shulman et al., 2006 [39]	17 healthy subjects	i-v −	Glutamate: brain medial prefrontal cortex, by photon magnetic resonance spectroscopy (MRI/MRS) vs. plasma (taken within 1 week), by HPLC/MS	No brain-plasma correlation
Huo et al., 2020 [46]	Subjects with and without Alzheimer’s disease (at time of death; N = 31 and 61, respectively)	p-m −	143 metabolites from five compound classes (amino acids, biogenic amines, acylcarnitines, glycerophospholipids, and sphingolipids): brain vs. serum, by ultra-high-pressure liquid chromatography (UPLC) tandem MS	No report on correlation
Wang et al., 2020 [47]	Alzheimer’s disease, mild cognitively impaired patients and unimpaired subjects (N = 92 total *, of whom AD N = 11)	p-m −	129 metabolites (the majority are not neurospecific): brain vs. serum, by gas chromatography time-of-flight mass spectrometry (GC-TOFMS)	No report on correlation
**Peptides**				
Chiaretti et al., 2004 [54]	9 children operated on for epilepsy	i-v +	BDNF, GDNF, NGF: brain (tissue surrounding epileptic lesions) vs. plasma, by ELISA	No report on correlation
Bharani et al., 2019 [48]	Subjects with Alzheimer’s disease and healthy controls (N = 22 total *)	p-m +	BDNF and pro-BDNF: brain (cortex Brodmann area 46, entorhinal cortex, hippocampus), by Emax ImmunoAssay and Western blot, respectively vs. serum, by ELISA and Western blot, respectively.	Significant hippocampus-serum correlation for pro-BDNF (***rho* = −0.43, *p* = 0.040**). No other correlations significant
Gadad et al., 2021 [6]	Subjects with mood disorder and healthy controls (N = 28 total *)	p-m +	BDNF, GDNF (and also IL-1b, IL-6): brain (Brodmann area 10) vs. plasma, by multiplex assay	Brain-plasma correlation: for IL-6 ***rho* = 0.44 (*p* = 0.031)**, for GDNF ***rho* = 0.37** (***p* = 0.05**, a trend), other—ns.
Ashton et al., 2019 [49]	Subjects with Alzheimer’s disease and healthy controls (N = 23 total *)	p-m −	NfL: brain (medial temporal gyrus), % density by immunostaining vs. plasma concentration measured serially (three times during 1–8 years prior to death), by Simoa method	Significant brain-blood correlation for NfL in blood sampled at the closest time to death **(*rho* = −0.47, *p* < 0.05)**
Bartolotti et al., 2016 [50]	32 subjects with Alzheimer’s disease and 33 cognitively unimpaired controls	p-m −	CREB, pCREB, and transcription cofactors—CREB-binding protein (CBP), p300: brain vs. PBMCs taken once within 5 years to death, by Western blot	Significant brain-PBMCs correlation for pCREB in a subgroup of AD patients whose blood was taken <3 years before death (*r* not reported, ***p* = 0.002**, N = 11). Other correlations—ns.
Buttarelli et al., 2009 [51]	11 subjects with Parkinson’s disease naive of dopaminergic drugs	i-v +	Dopamine transporter: brain (caudate and putamen nuclei of the striatum), by SPECT (^123^I-fluopane binding) vs. peripheral blood lymphocytes, by immunocytochemistry	No significant correlations
Kanegawa et al., 2016 [52]	31 healthy subjects	i-v +	TPRO: brain (highly expressed in microglia and macrophages) vs. circulating blood cells, by PET [^11^C]PBR28 binding, twice within a year	Significant brain-blood correlation at both first (***r* = 0.85**, N = 31) and second (***r* = 0.90**, N = 25) measurements (***p* < 1 × 10^8^**) and for the changes (***r* = 0.60, *p* = 0.002**).
Obukhova et al., 2021 [53]	28 patients with glioma	i-v +	Acetylcholinesterase: glioma tissue (per 1 g of protein) vs. whole blood (per 0.1 g of hemoglobin), by photo colorimetric analysis	“Highly” significant brain-blood correlation ***rho* = 0.63**

^ Conditions: p-m—post-mortem, i-v—in vivo brain study; “+”—brain and periphery data were collected on the same day, “−”—not on the same day. * The number per group was not reported. The bold emphasizes findings with significant correlation.

### 3.3. Discussion

For this review, we found only 13 human studies, which informed on brain-periphery correlations for neurospecific molecules. The results were rather contradictory; four of these studies did not reveal any significant correlation. The studies were heterogeneous with respect to the measured analytes: only four analytes were examined in more than one study (phenylalanine, glutamine, glutamate, and BDNF); each substance was examined in two studies. For the first three molecules (amino acids), the result was negative in one study and positive in another. The discrepancies may be related to the technique for the specimen measurement (biochemical vs. neuroimaging) or to the sampling condition (same vs. not the same day for the brain and periphery data acquisition). For BDNF, both studies showed no correlation between the brain and serum/plasma; both were similar by conditions (post-mortem, same day data acquisition) and measuring techniques (biochemical analysis). These BDNF results were in accordance with our preliminary in vivo finding on the absence of correlation for BDNF between the hippocampus and serum or leukocytes in 20 epileptic patients who underwent brain surgery [55]. Nevertheless, we found a correlation for NSE between the hippocampus and leukocytes (Figure 2).

Two other substances from the reviewed studies should be noted separately because of the very strong brain-periphery correlation found for them. One was plasma free tryptophan (***r* = 0.97**) and another one was TPRO in blood cells (***r* = 0.90**) (Figure 3). Such strong correlation suggests that, even if analyzed in a single study, the analyte could indeed serve as a true peripheral neuromarker. Both studies were in vivo, the data acquisition was performed in one day, and measurement techniques were the same for the brain and periphery. The approach of measuring and comparing the analyte levels in the biosamples of different tissues taken in vivo and simultaneously seems to be the most accurate. However, the availability of such samples is limited by certain forms of a few pathologies for which brain surgery is recommended (e.g., focal drug-resistant epilepsy), and the number for that type of surgery is relatively small. For example, approximately 1.5% of people newly diagnosed with epilepsy may undergo epilepsy surgery [56]. On the other hand, samples of more frequent surgery for brain oncology should be considered with caution, as brain tumors differ significantly from the healthy brain tissue in qualitative and quantitative characteristics. Moreover, a recent proteome study reported that cancer-related proteins were detected in the healthy zone (fluorescence negative tumor periphery) in the brains of patients with glioblastoma [57].

In this regard, the approach of advanced neuroimaging techniques, such as MRI/MRS, PET, SPECT, appears to have great prospects, as these methods can be applied in a wide range of normal and pathological conditions. The field of neuroimaging research is rapidly developing, and we may expect new breakthrough findings on the matter of brain-periphery concordance with the increasing accessibility of MRI/MRS and PET/SPECT and the emergence of novel radioactive tracers. The study by Kanegawa et al. (2016) [52] on TPRO represents a good example of the trend.

Although gene expression and DNA methylation patterns were out of scope of the review, attempts at research on the concordance of these parameters between the brain and periphery are currently actively developing. Despite the absence of significant correlation between the brain and periphery while analyzing the methylomic profiling of whole blood versus brain tissue, for some genes a strong correlation across the tissues was found. For example, high concordance in methylation patterns of *DRD4* gene encoding the dopamine receptor D4 [60,61], or *DAT1* gene encoding the dopamine transporter [62] between blood and brain tissue was found in humans. Moreover, a paper by Shumay et al. (2012) [63] reported on the robust association of the methylation of a promoter of *MAOA* gene, encoding monoamine oxidase A, in blood cells with brain levels of the enzyme.

We should also highlight the emerging role of exosomes being the promising candidates bridging the gap between the brain and periphery. Exosomes are extracellular vesicles of 50–150 nm in diameter that are released by cells; they play an important role in intercellular communication and, thus, they circulate freely in body fluids, enter target cells, and may easily cross multiple anatomical and physiological boundaries, including the highly selective BBB [64]. They carry diverse molecular cargoes, such as nucleic acids (DNA, RNA, ncRNA, miRNA), lipids, and proteins that have a footprint reflective of their parental origin (both parent cells and extracellular environment) [65,66]. Compared to other substrates, the discovery of brain exosomes has been delayed. Conclusive demonstration of brain extracellular vesicles was first reported at the beginning of the last decade. Since then, the research in the field of brain exosomes has increased progressively [64]. The ability of exosomes from the CNS to readily cross the BBB has generated enthusiasm in their investigation as a potential biomarker source. Exosomes are considered to mirror heterogeneous biological changes that occur during the progression of neuropsychiatric diseases [67]. Animal and human studies demonstrated an accuracy of several types of exosomes and miRNAs in detecting mild, moderate, and severe traumatic brain injury [68]. Moreover, as a novel class of circulating biomarkers, exosomes can provide new insights on the biophysical and/or biochemical properties of biomolecules of brain origin. New technological advances are developing to overcome existing technical challenges in measuring rare exosome biomarkers and to reveal novel features of these biomarkers for accurate, blood-based detection of brain pathology [69].

The limitations met during the review process included a substantial heterogeneity across studies in neurospecific substances analyzed, as well as in methods used to measure them. Another limitation was related to risk of bias due to missing results, as some of the publications did not report on correlation, although they matched the eligibility criteria.

## 4. Conclusions

Quite a few studies have investigated the correlation for neurospecific molecules between the brain and periphery (blood) in humans. The studied molecules were different, and only for three of them was there any—and yet weak—evidence. BDNF in plasma or serum does not reflect brain BDNF content, whereas free tryptophan (in plasma) and TPRO (in blood cells) can serve as true peripheral biomarkers. In the next few years, we expect the emergence of studies elucidating the brain-periphery concordance for a wide range of neurospecific molecules with advanced in vivo metabolomic analyses, neuroimaging techniques, and blood assays for exosomes of brain origin.

## Figures and Tables

**Figure 1 ijms-23-09193-f001:**
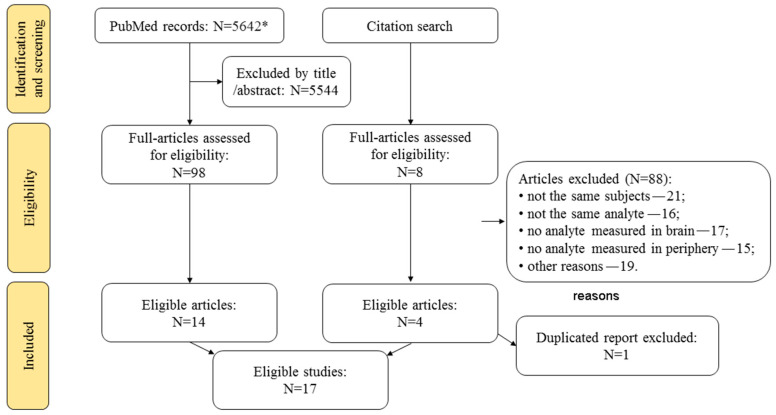
Search of the published studies eligible for the systematic review. * Includes publications of interest already known to the authors.

**Figure 2 ijms-23-09193-f002:**
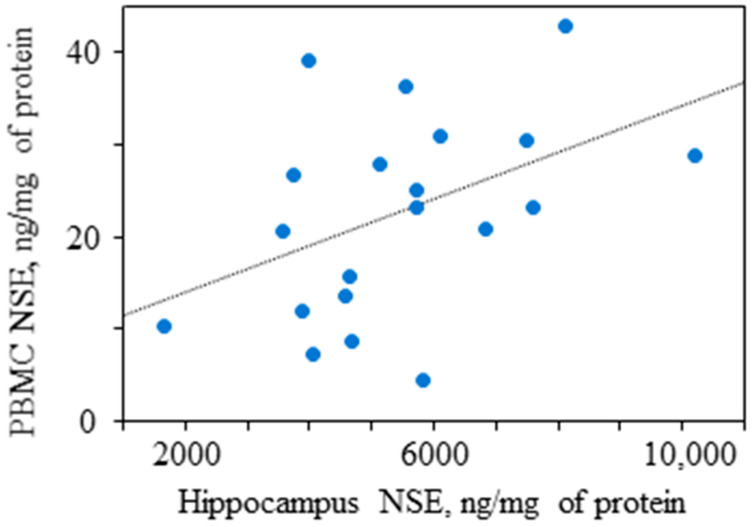
Correlation for concentration of NSE (corrected by total protein) between the hippocampus and PBMCs in 20 operated epileptic patients (***r* = 0.45, *p* = 0.047**, Pearson test). NSE concentration was measured using ELISA with Vector-Best kits (Novosibirsk, Russia). The regression line is shown.

**Figure 3 ijms-23-09193-f003:**
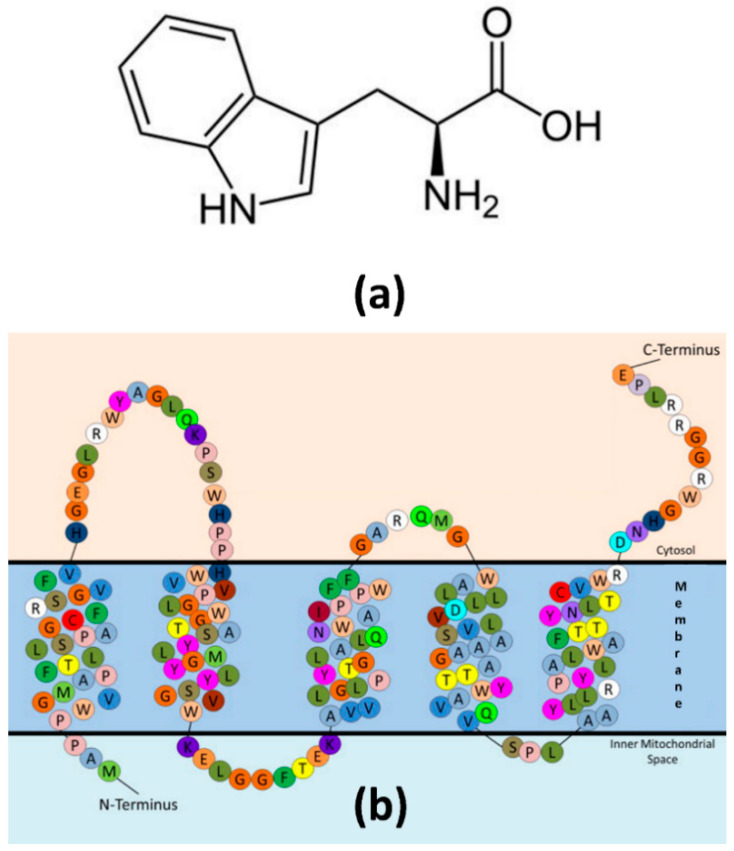
Neurospecific molecules with strong brain-periphery correlation. (**a**) Tryptophan. (**b**) Topological model of human TSPO reproduced from [58], Copyright © 2013, with permission from Elsevier and corresponding author Professor Louis M. Rendina. The figure of TSPO in [58] was adapted from [59].

## Data Availability

The data presented in this study are available on request from the corresponding author.

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
