# Peer review of "Neurospecific Molecules Measured in Periphery in Humans: How Do They Correlate with the Brain Levels? A Systematic Review"

_ijms, 2022, doi:10.3390/ijms23169193_

Round 1

Reviewer 1 Report

This review "Neurospecific Molecules Measured in Periphery in Humans: How Do They Correlate with the Brain Levels? A Systematic Review" by Maria A Tikhonova et .al has been written well, the content is good and have potential to publish in the current form. I strongly recommend this paper to be accepted for publication.

Author Response

Dear Reviewer, we would like to cordially thank you for the careful review of our manuscript. We greatly appreciate your high esteem of our attempt to provide a comprehensive systematic review of the current knowledge about correlation of neurospecific molecules in the brain and periphery.

Reviewer 2 Report

Manuscript ID: ijms-1852878

Manuscript Title: Neurospecific Molecules Measured in Periphery in Humans: How Do They Correlate with the Brain Levels? A Systematic Review

Dear editor,

This is a systematic review of studies addressing an important problem that whether quantification of peripheral biomarkers represents their levels in the central nervous system. According to their findings, BDNF in plasma or serum does not reflect brain content, whereas free tryptophan (in plasma) and translocator protein (in blood cells) can serve as peripheral biomarkers. These findings are of importance in terms of the interpretation of clinical biomarkers studies, particularly relevant to neuropsychiatric research. Therefore, I believe the paper is valuable. The authors have sufficiently emphasized the role of BBB. Despite the intro being well organized, the rationale and aims of the study should be given in the last section of the introduction. Since it is a systematic review, the overall risk of bias in the publications should be stated. The time frame of the searched publications (from which date to 2022) should be given. Each review has limitations and should be mentioned. 

Author Response

Dear Reviewer, we would like to cordially thank you for the careful review of our manuscript and for your valuable comments and suggestions. We greatly appreciate your high esteem of our attempt to provide a comprehensive systematic review of the current knowledge about correlation of neurospecific molecules in the brain and periphery. We have thoroughly revised the text considering all the comments, which helped us to improve the manuscript. All major corrections made in the text are highlighted with yellow color. We believe that the revised version would be more clear and interesting for the readership of the journal.

- Despite the intro being well organized, the rationale and aims of the study should be given in the last section of the introduction.

Done. We added a section 1.6 Study Objectives.

- Since it is a systematic review, the overall risk of bias in the publications should be stated. Each review has limitations and should be mentioned.

- Done. The review has two main limitations. One limitation is the increased heterogeneity across studies in neurospecific analytes as well as methods applied to measure them. Another limitation is related to risk of bias due to missing results, as some of the publications did not report on correlation although they matched the eligibility criteria. We mentioned these limitations in lines 431-435.

- The time frame of the searched publications (from which date to 2022) should be given.

The time frame (October 1970 - June 2022) was specified in lines 218-219.

Reviewer 3 Report

This review is devoted to neurospecific molecules. The article fits the theme of the magazine and the theme of the special issue. The relevance and necessity of such a review is beyond doubt. However, there are some points that I recommend eliminating:

1. It is desirable to make the introduction more complete. Indicate if anyone has previously made reviews on this topic and on related topics, why it became necessary to write just such a review by the authors.

2. Conclusions are an important element of a scientific study (even a review). Please consider this.

3. Adding pictures could qualitatively decorate this review. Insert at least pictures of some molecules.

4. It is desirable to slightly expand the list of references.

5. Where appropriate, please cite the following work: 10.3390/ijms23031602.

Author Response

Dear Reviewer, we would like to cordially thank you for the careful review of our manuscript and for your valuable comments and suggestions. We greatly appreciate your high esteem of our attempt to provide a comprehensive systematic review of the current knowledge about correlation of neurospecific molecules in the brain and periphery. We have thoroughly revised the text considering all the comments, which helped us to improve the manuscript. All major corrections made in the text are highlighted with yellow color. We believe that the revised version would be more clear and interesting for the readership of the journal.

  1. It is desirable to make the introduction more complete. Indicate if anyone has previously made reviews on this topic and on related topics, why it became necessary to write just such a review by the authors.

No one has previously made a review in the topic. The necessity of such review is discussed in manuscript lines 32-42.

  1. Conclusions are an important element of a scientific study (even a review). Please consider this.

The last paragraph of the manuscript is currently titled as "5. Conclusions".

  1. Adding pictures could qualitatively decorate this review. Insert at least pictures of some molecules.

We added Figure 3 with images of neurospecific molecules with strong brain-periphery correlation.

  1. It is desirable to slightly expand the list of references.

We followed the PRISMA guidelines for systematic reviews and included only relevant references to the Results according to the determined criteria. The list of references was expanded with references related to closely relevant topics discussed in the paper, including the main approaches in this research field along with the difficulties, challenges, and prospects. In the revised version, we added two novel references (37, 58).

  1. Where appropriate, please cite the following work: 10.3390/ijms23031602.

Unfortunately, the above-mentioned work does not fit the theme of our systematic review. We cannot cite it in the paper.